# How valuable are the questions and answers generated by large language models in oral and maxillofacial surgery?

**Kyuhyung Kim[1]☉, Sae Byeol Mun[2]☉, Young Jae Kim[3], Bong Chul Kim [1]\*, Kwang Gi Kim [4,5]\***

**1** Department of Oral and Maxillofacial Surgery, Daejeon Dental Hospital, Wonkwang University College of Dentistry, Daejeon, Republic of Korea, **2** Department of Health Sciences and Technology, GAIHST, Gachon University, Incheon, Republic of Korea, **3** Gachon Biomedical & Convergence Institute, Gachon University Gil Medical Center, Incheon, Republic of Korea, **4** Department of Biomedical Engineering, College of IT Convergence, Gachon University, Seongnam, Republic of Korea, **5** KMAIN, Seongnam, Republic of Korea

☉ These authors contributed equally to this work.
\* bck@wku.ac.kr (BCK); kimkg@gachon.ac.kr (KGK)

## Abstract

### Introduction

In this study, we aim to evaluate the ability of large language models (LLM) to generate questions and answers in oral and maxillofacial surgery.

### Methods

ChatGPT4, ChatGPT4o, and Claude3-Opus were evaluated in this study. Each LLM was instructed to generate 50 questions about oral and maxillofacial surgery. Three LLMs were asked to answer the generated 150 questions.

### Results

All 150 questions generated by the three LLMs were related to oral and maxillofacial surgery. Each model exhibited a correct answer rate of over 90%. None of the three models were able to answer correctly all the questions they generated themselves. The correct answer rate was 97.0% for questions with figures, significantly higher than the 88.9% rate for questions without figures. The analysis of problem-solving by the three LLMs showed that each model generally inferred answers with high accuracy, and there were few logical errors that could be considered controversial. Additionally, all three scored above 88% for the fidelity of their explanations.

### Conclusion

This study demonstrates that while LLMs like ChatGPT4, ChatGPT4o, and Claude3-Opus exhibit robust capabilities in generating and solving oral and maxillofacial

**Data availability statement:** All relevant data are within the paper and its Supporting Information files.

**Funding:** This work was supported by the National Research Foundation of Korea (RS-2024-00451221, Pf. Bong Chul Kim), the Development Fund Program of Wonkwang University College of Dentistry funded by Dentium Co., Ltd., 2024, (Pf. Bong Chul Kim), and the GRRC program of Gyeonggi province (GRRC-Gachon 2023 (B01), Development of AI-based medical imaging technology, Kwang Gi Kim). The funding bodies had no role in the design of the study, data collection, analysis, interpretation of data, or writing of the manuscript.

**Competing interests:** The authors have declared that no competing interests exist.

surgery questions, their performance is not without limitations. None of the models were able to answer correctly all the questions they generated themselves, highlighting persistent challenges such as AI hallucinations and contextual understanding gaps. The results also emphasize the importance of multimodal inputs, as questions with annotated images achieved higher accuracy rates compared to text-only prompts. Despite these shortcomings, the LLMs showed significant promise in problem-solving, logical consistency, and response fidelity, particularly in structured or numerical contexts.

## Introduction

Since AI was first defined in 1956, it has evolved significantly, with key milestones such as AlphaGo's victory over professional Go players in 2016 and the release of ChatGPT in 2022, which marked the rise of large language models (LLMs) based on transformer neural networks [1]. These models, including OpenAI's ChatGPT, Anthropic's Claude, Google's PaLM 2, and Meta's LLaMa 2, have become widely accessible through subscriptions or open-source platforms, helping to advance AI research and usage [2]. Notably, image recognition capabilities, which are critical in medical and dental fields, were introduced in newer versions like ChatGPT4 and Claude-3 [3,4].

Studies have extensively examined LLM performance in education and healthcare. For example, research on ChatGPT3.5 showed that it performed well in UK and German examination settings, though limitations such as data bias and outdated information were noted [5,6]. In healthcare, LLMs have been shown to assist in clinical decision-making and diagnosis, improving processes in areas like radiotherapy planning and surgical planning in dentistry [7–10]. Despite these promising results, concerns about bias, outdated data, and patient privacy persist, particularly in specialties like orthopedics and maxillofacial surgery [10,11].

In medical education, LLMs have been tested on professional licensing exams. ChatGPT3.5 and its successor, ChatGPT4, demonstrated strong results, meeting passing scores in both US and UK dental licensing exams [12,13]. ChatGPT4 showed improved performance over earlier versions, suggesting its potential as a valuable tool in medical training [14]. Ongoing comparisons between ChatGPT4 and Claude3-Opus show similar overall performance, though variations exist across different fields, indicating the need for further evaluation of these models' strengths and weaknesses in specific domains [15].

However, most of the LLM studies in various medical specialties were focused on measuring the ability to solve human-made questions. Therefore, in this study, to accurately assess the performance of LLMs, we aim to evaluate their ability to generate answers and questions in our specialty areas of oral and maxillofacial surgery.

## Materials and methods

This article does not contain any studies with human participants or animals performed by any of the authors. No ethical committee approval is required since this study is performed on publicly available open-source data. For this type of study, formal consent is not required.

This study evaluated three LLMs: ChatGPT4, ChatGPT4o, and Claude3-Opus. The models were tested using updates from May 16, 2024 (ChatGPT4 and ChatGPT4o), and February 06, 2024 (Claude3-Opus). This study was conducted between June 12, 2024, and July 10, 2024, at Wonkwang and Gachon University, Republic of Korea.

**Question generation and evaluation**

Three LLMs were prompted with the following request (Fig 1):

Create 50 questions for the Oral and Maxillofacial Surgery Specialist Exam to find 1 correct answer out of 5 options and provide the correct answer for each question and the basis for the correct answer.

Each question must meet one or more of the following:

1. Simple knowledge measurement

2. Causal inference

3. Example through patient case

4. Includes schematic diagram, photography, X-ray, CT and/or MRI

**Expert review process**

Two oral and maxillofacial surgeons assessed the generated questions. These specialists were selected based on their expertise in oral and maxillofacial surgery and their prior experience with standardized examination question development. The review process involved:

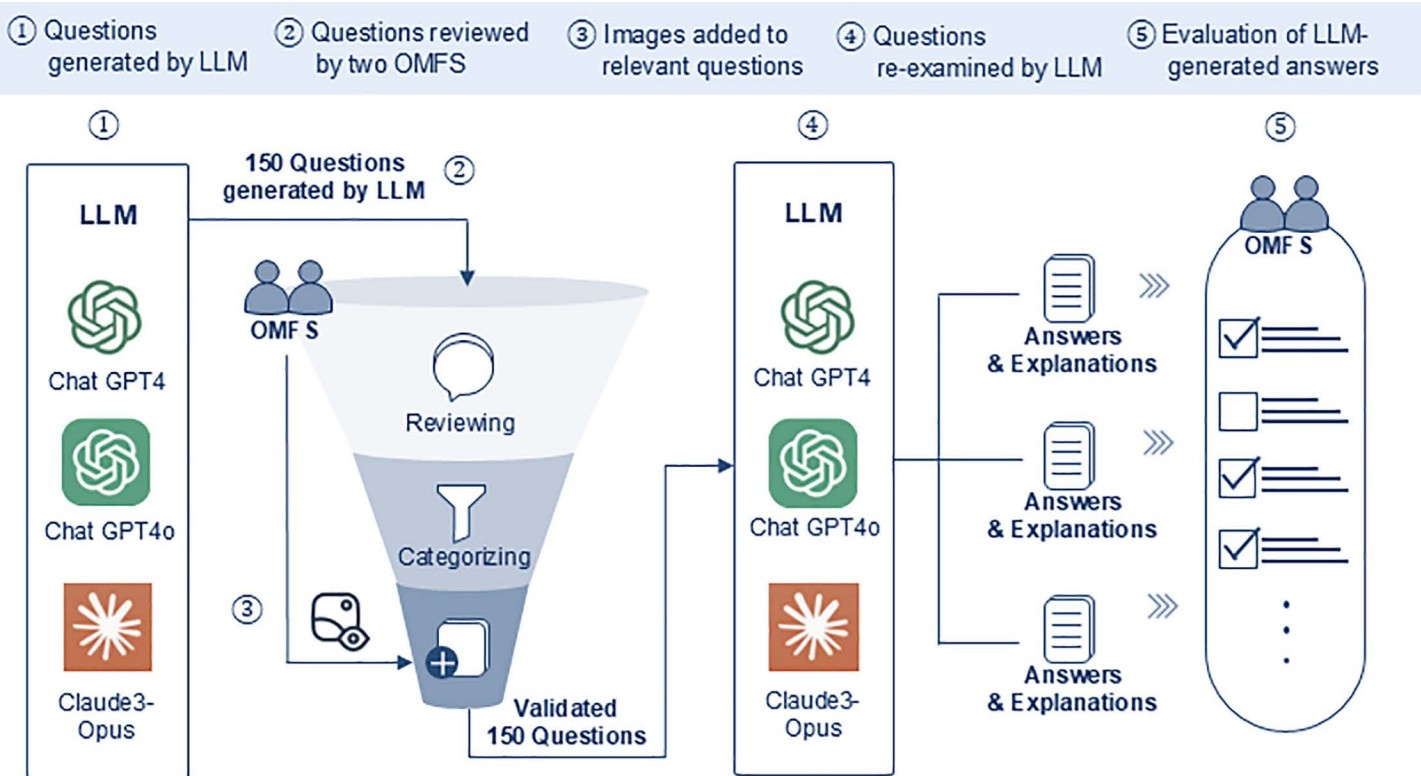

**Fig 1. A flow diagram showing the process by which large language models generate and answer questions about oral and maxillofacial surgery in this study.** LLM: large language models, OMFS: oral and maxillofacial surgeon.

- Verifying compliance with predefined criteria (knowledge measurement, causal inference, patient case-based scenarios, and image inclusion).

- Assessing clinical relevance, alignment with standardized exam formats, and logical coherence of the provided explanations.

- Evaluating the accuracy of answers, completeness of problem-solving approaches, and consistency of logical reasoning.

- Categorizing questions into subfields such as dentoalveolar, trauma, infection, and cysts.

### Image integration and modification

For questions requiring images, open-source images from Google were incorporated, ensuring they were sourced only from credible institutions. Image utilization per model was as follows: ChatGPT-4 (9 questions), ChatGPT-4o (18 questions), and Claude 3-Opus (18 questions). When only a specific part of an image needed to be evaluated, relevant areas were highlighted using arrows or circles for clarity (Fig 2). To assess LLMs' ability to generate independent explanations, the provided answers and rationales were removed before resubmitting the questions to the models.

### Performance evaluation and statistical analysis

The LLMs' capabilities were assessed based on:

- Logical Accuracy and Thoroughness: If an explanation contained correct isolated facts but lacked logical relevance to the problem, it was deemed incorrect. Similarly, a correct solution with an explanation consisting only of disjointed facts was considered insufficient.

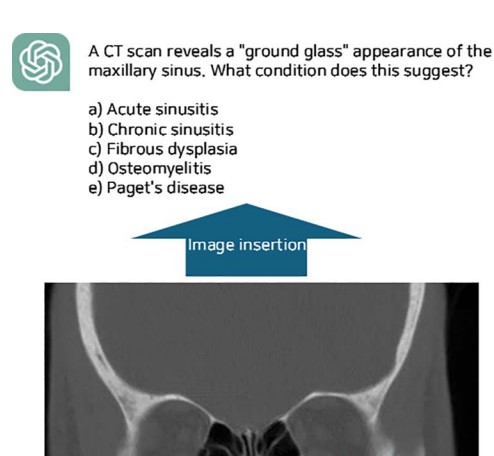

**Fig 2. An example of inserting images verified and labeled by experts from credible institutions among open sources into questions generated by large language models.**

- Quantitative Assessment: Responses were classified as correct or incorrect, and explanations were labeled as detailed or insufficient. Image interpretation abilities were also evaluated, and cross-verifications were performed to ensure logical consistency.

- Reliability Testing: Intra-rater and inter-rater reliability were assessed for all 150 questions. Each expert evaluated all questions twice at a one-week interval, and a final gold-standard evaluation was established based on consensus.

- Statistical Analysis: An independent t-test was conducted to compare performance metrics among the three LLMs. Accuracy, precision, recall, and F1-scores were calculated using Scikit-learn version 1.4.2. The statistical analyses assumed normality in data distribution and independence between observations. Adjustments for multiple comparisons were considered unnecessary as the analyses were hypothesis-driven and controlled.

## Results

The 150 questions generated by the three LLM models were all related to oral and maxillofacial surgery. There were no intra- or inter-rater errors. Two oral and maxillofacial surgeons independently evaluated and solved all 150 questions at one-week intervals. There was no difference in the accuracy of the answers, and it was found that both experts completely agreed.

The analysis of the categories for the 50 questions generated by each model in the oral and maxillofacial surgery field is shown in Fig 3. Each question was categorized into a total of 15 categories; ChatGPT4 generated the most questions in the trauma category, ChatGPT4o in the benign tumor and lesion category, and Claude3-Opus in the salivary gland category. Consequently, ChatGPT4 showed the most balanced distribution across categories, followed by ChatGPT4o and Claude3-Opus.

The analysis of the response rate of each model showed that all exceeded 90% (Table 1). All three LLMs showed a response rate over 90%, with the highest rates in the following order: Claude3-Opus (92.7%), ChatGPT4 (91.3%), ChatGPT4o (90.0%). The correct answer rate for questions generated by ChatGPT4o and Claude3-Opus was 93.3%, while that for ChatGPT4 was 87.3%. It was observed that none of the models achieved a 100% response rate for their own questions.

A confusion matrix was calculated using the ground truth (initial labeling performed by the LLMs during question generation, with subsequent validation by the two experts) for each model's questions and responses (Fig 4). Accuracy, Precision, Recall, and F1-Score were calculated and validated, and the quantitative evaluation results are shown in Table

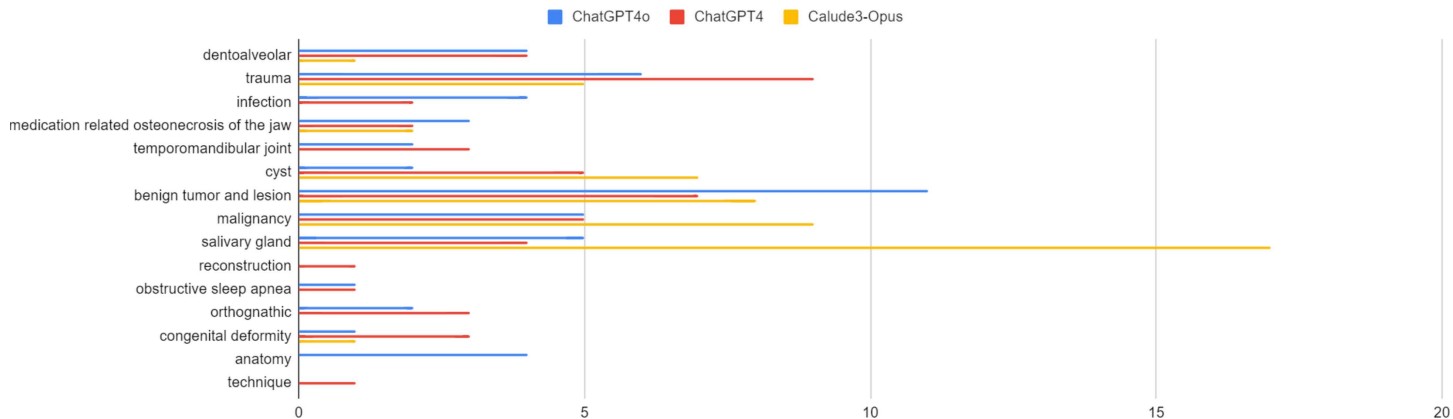

**Fig 3. Distribution of question generation areas in large language models.**

2. Based on the questions generated by each model, on average, the highest performance was observed when solving questions created by ChatGPT4o, followed by Claude3-Opus and ChatGPT4.

   Based on the responses of each model, on average, the responses created by ChatGPT4 showed the highest performance, followed by Claude3-Opus, and ChatGPT4o. The response rate for questions with images was 97.0%, higher than 88.9% for questions without images (Fig 5 and Table 3). In the case of ChatGPT4, the response rate for questions with images was 91.1%, slightly lower than the 91.4% for questions without images. However, ChatGPT4o and Claude3-Opus correctly answered 100% of the questions with images, significantly exceeding the response rate for questions without images (85.7% for ChatGPT4o, 89.5% for Claude3-Opus). The responses of each model all showed statistically significant results ($p < 0.001$). The analysis of problem-solving by the three LLMs showed that each model generally inferred answers with high accuracy, and there were few logical errors that could be considered controversial (Table 4). The fidelity of explanation was 96.0% for Claude3-Opus, 95.3% for ChatGPT4, and 88.7% for ChatGPT4o (Fig 6 and Table 5).

## Discussion

Existing research using LLMs has typically assigned each model a single, discrete role. AI models have been tasked with analyzing scenarios and suggesting methods [8–10] or solving pre-existing problems [4–6,12]. Since LLMs are programs created through algorithmic code, they are likely to perform well in "logical analysis", that is, in analyzing scenarios or solving problems. According to Sun et al., the DetermLR framework employs premise identification, premise prioritization, and an iterative reasoning process to enable LLM-based reasoning to closely mirror human cognitive reasoning. This framework enhances logical reasoning by identifying and categorizing premises, prioritizing them based on their relevance to the conclusion, and iteratively refining the reasoning process. We propose that evaluation should include 'the creation of situations or problems'. Unlike previous research, our study aims for the model to play a bidirectional role by both presenting and solving problems.

   ChatGPT4, developed by OpenAI, is a model renowned for its high performance in natural language processing (NLP) and generation tasks[3]. Compared to its predecessors, this model has been trained on a larger dataset and undergone a longer training process, significantly enhancing its text comprehension and generation capabilities. As a versatile language model, ChatGPT4 can be employed in various tasks, including conversation, translation, summarization, and question answering, offering high accuracy and flexibility. ChatGPT4o, OpenAI's latest version of their language model, excels in a wide array of text generation and understanding tasks. Trained on an extensive amount of text data, it is adept at answering questions on diverse topics, assisting with creative writing, and providing explanations for technical issues. Incorporating the latest AI advancements, ChatGPT4o facilitates more natural and fluent conversations. Claude3-Opus, developed by Anthropic, is an advanced language model emphasizing human-centric AI research and safety. It demonstrates exceptional performance in natural language understanding and generation, designed with a focus on ethical and safe AI use. Claude3-Opus can be utilized for answering complex questions, generating conversations, and analyzing text, with a strong emphasis on accurately interpreting user intent and context. While each of these models exhibits slight

**Table 1. Score of each large language model.**

|  | Generated by ChatGPT4 (n = 50) | Generated by ChatGPT4o (n = 50) | Generated by Claude3-Opus (n = 50) | Total (response rate) |
|---|---|---|---|---|
| Answered by ChatGPT4 | 44 (88.0%) | 46 (92.0%) | 47 (94.0%) | 137 (91.3%) |
| Answered by ChatGPT4o | 44 (88.0%) | 47 (94.0%) | 44 (88.0%) | 135 (90.0%) |
| Answered by Claude3-Opus | 43 (86.0%) | 47 (94.0%) | 49 (98.0%) | 139 (92.7%) |
| Total (response rate) | 131 (87.3%) | 140 (93.3%) | 140 (93.3%) |  |

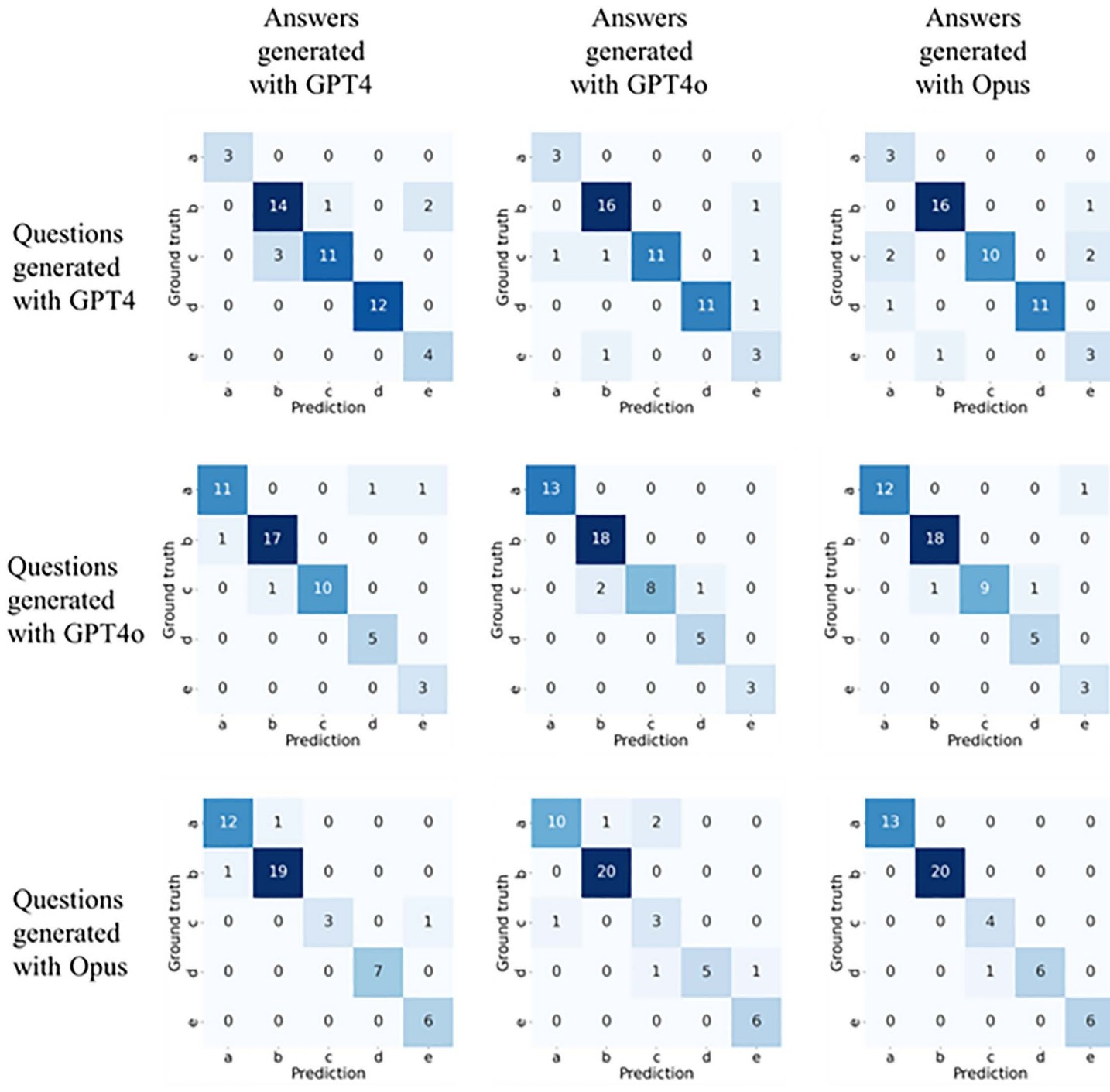

**Fig 4. Comparative heatmap of response results from ChatGPT4, ChatGPT4o, and Claude3-Opus.**

**Table 2. Performance metrics for the output of each LLM model.**

| Questions | Answers | Accuracy | Precision | Recall | F1-Score |
|---|---|---|---|---|---|
| | ChatGPT4 | 88 | 88.1 | 92.2 | 89.4 |
| ChatGPT4 | ChatGPT4o | 88 | 82.8 | 87.9 | 84.2 |
| | Claude3-Opus | 86 | 78.8 | 86.4 | 80 |
| | ChatGPT4 | 92 | 88.9 | 94 | 90.9 |
| ChatGPT4o | ChatGPT4o | 94 | 94.7 | 94.5 | 94 |
| | Claude3-Opus | 94 | 90.6 | 94.8 | 92 |
| | ChatGPT4 | 94 | 94.6 | 92.5 | 93.1 |
| Claude3-Opus | ChatGPT4o | 88 | 84.4 | 84.7 | 83.3 |
| | Claude3-Opus | 98 | 96 | 97.1 | 96.2 |

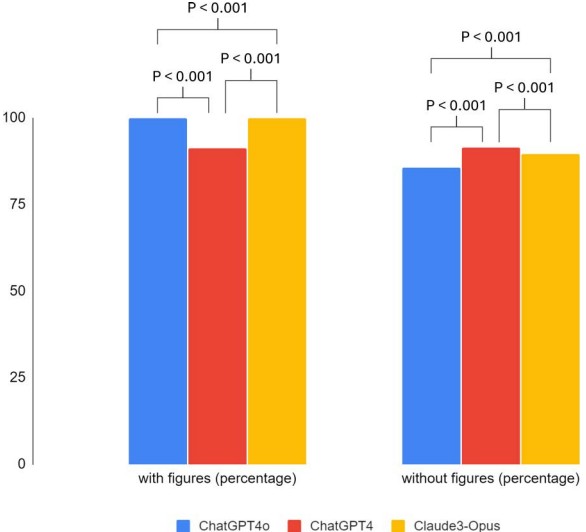

**Fig 5. Correct response rate according to figures.**

**Table 3. Correct response rate according to figures.**

| | ChatGPT4 | ChatGPT4o | Claude3-Opus | Mean |
|---|---|---|---|---|
| With figures (score, n=45) | 41 | 45 | 45 | 43.7 |
| With figures (response rate) | 91.1% | 100% | 100% | 97.0% |
| Without figures (score, n=105) | 96 | 90 | 94 | 93.3 |
| without figures (response rate) | 91.4% | 85.7% | 89.5% | 88.9% |

differences based on the databases used for training by their respective developers, all possess sophisticated NLP capabilities and serve as powerful tools applicable across various domains [15].

Upon comparing and analyzing the results of the LLM models, it was observed that problems involving numerical answers had a higher accuracy rate compared to those without numbers. This increased accuracy can be attributed to the application of clear mathematical or data-based rules when deriving numerical answers. Numbers provide specific and unambiguous information, making it easier for the models to learn patterns related to numerical problem-solving.

**Table 4. Consistency of logic.**

|  | ChatGPT4 | ChatGPT4o | Claude3-Opus |
|---|---|---|---|
| answer T & basis T | 137 | 135 | 139 |
| answer T & basis F | 0 | 0 | 0 |
| answer F & basis T | 0 | 0 | 0 |
| answer F & basis F | 13 | 15 | 11 |
| Total | 150 | 150 | 150 |

T: True, F: False.

answer: the answer generated by each model for the input question.

basis: description of the basis on which each model generated the answer.

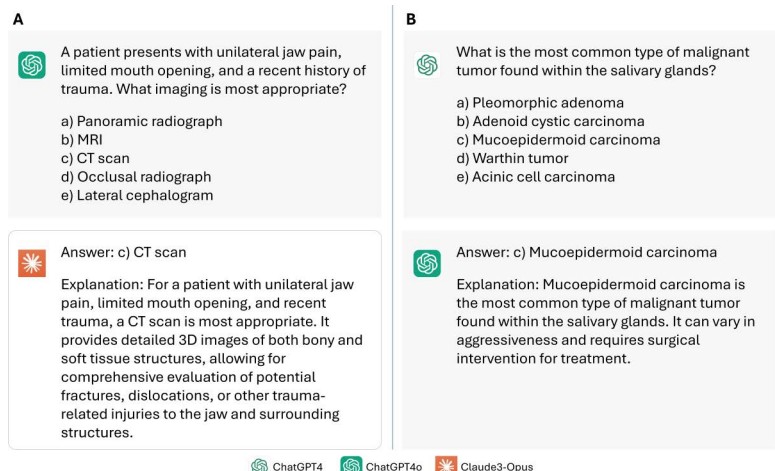

**Fig 6. Examples of A. when the explanation is detailed and B. when it is insufficient.**

**Table 5. degree of faithfulness to the explanation.**

|  | ChatGPT4o | ChatGPT4 | Claude3-Opus |
|---|---|---|---|
| Detailed (response rate) | 88.7 | 95.3 | 96.0 |
| Insufficient (response rate) | 11.3 | 4.7 | 4.0 |

Conversely, problems without numerical components tend to be more subjective or dependent on context and meaning, making it more challenging and ambiguous to derive correct answers. These problems often require extensive back-ground knowledge, logical reasoning, and contextual understanding, elements that can be difficult for models to accurately grasp and apply. Therefore, the higher accuracy rate in numerical problems can be attributed to the clarity and regularity that numbers provide, enabling the models to understand and solve these problems more precisely.

In this study, we presented several conditions when each LLM generated questions. The "simple knowledge measurement" was intended to measure each model's understanding or recall of basic concepts, facts, or information. "Causal inference" was to confirm the LLM's process of determining whether and how one variable directly influences another through evidence, reasoning, or statistical analysis. "Example through patient case" was intended to confirm LLMs' ability

to create a detailed narrative or scenario describing a specific patient's condition, history, and treatment to illustrate medical principles or practices. And "includes schematic diagram, photography, X-ray, CT, and/or MRI" was to evaluate the LLMs' understanding of anatomical or physiological features through medical image data. As a result, we enabled LLMs to generate questions appropriate to the purpose of this study.

In this study, each LLM was fed with images that included necessary annotations and highlights. The accuracy rates of the LLMs were higher on problems where images were provided compared to those without images. This aligns with other research indicating that the accuracy of LLMs improves when multiple forms of evidence are provided.

The three LLM models, ChatGPT4, ChatGPT4o, and Claude3-Opus, were tasked with generating 50 oral and maxillofacial surgery questions along with their answers and explanations based on their training data. However, none of the models could provide 100% accurate answers to the 50 questions they generated themselves. This reflects the persistent issue of AI hallucinations, which may arise due to several contributing factors. The causes of such AI hallucinations are known to include data quality issues, overfitting, and limited model understanding [16].

In this research, oral and maxillofacial surgery is highly specialized, requiring up-to-date information and in-depth knowledge. Although LLMs are trained on extensive datasets, they might not encompass the most current and specific medical knowledge. Additionally, LLMs generate text based on probabilities and patterns within the data, potentially leading to an incomplete understanding of complex medical concepts and missing subtle nuances that human experts can capture. Furthermore, the knowledge of LLMs is constrained to the data available up to a certain point, which means recent advancements may not be reflected. Misinterpretation or mixing of information can lead to inaccurate responses, and the lack of practical experience implies LLMs lack the insights that can only be gained through hands-on practice. The generated content may also be ambiguous or vague, making it challenging to select a clear and definitive answer. Moreover, the basis of the responses is not consistently scientifically accurate or relevant. Biases or inaccuracies inherent in the training data may also be reflected in the outputs of the LLMs. In oral and maxillofacial surgery, one way to mitigate AI hallucination issues when using LLMs is through the 'Human-in-the-Loop (HITL)' approach. This can involve either participating in the development process or providing real-time feedback on the output generated by the LLMs [16]. In research and educational contexts, engaging in the development process can be particularly effective, whereas, in clinical settings, real-time feedback is likely to yield quicker results.

One limitation of this study is its reliance on evaluating LLM performance using a specific set of oral and maxillofacial surgery questions, which may not generalize to broader medical or surgical contexts. By generating 50 questions each, a total of 150 questions were created by the LLMs, and solving all 150 questions by each LLM resulted in 450 responses. According to IZ Sadiq and Richard, in complex fields such as oral and maxillofacial surgery, a sample size of 100–300 questions is typically required [17]. While the sample size in this study falls within that range, the complexity of all questions may not fully reflect the field. Additionally, LLMs might have produced similar questions due to coding similarities, suggesting that a larger sample size may have been needed. Additionally, the models' reliance on open-source images from Google could introduce variability in image quality and relevance, potentially affecting the accuracy of responses. The study also does not explore the long-term adaptability of LLMs to evolving medical knowledge.

Another limitation of this study is its reliance on AI-generated content, which introduces challenges in ensuring the real-world applicability of the findings. The generated questions and answers may not fully align with the complexities and nuances of clinical and educational environments, where human expertise and contextual judgment play critical roles. Additionally, AI models can exhibit limitations such as hallucinations or biases, which may reduce the reliability of their output in high-stakes applications like patient care or medical training. These factors underscore the need for cautious integration of LLMs into clinical and educational settings, with robust validation processes and continuous oversight by

human experts. Future research could expand the scope to include a wider range of surgical specialties, evaluate real-world clinical scenarios to test the practical relevance of LLM outputs, use standardized medical images, and assess the models' ability to incorporate real-time updates in medical practice. Evaluating patient-centered decision-making capabilities and ethical considerations related to AI-generated content could also provide valuable insights.

The deployment of LLMs in clinical and educational contexts may raise ethical concerns, including risks of data bias, patient privacy violations, and over-reliance on AI in decision-making. These risks could lead to inaccurate diagnoses or compromised patient care, especially if models produce hallucinations or fail to contextualize medical complexities. Safeguards should include a HITL framework to validate outputs, periodic audits of model performance against updated medical standards, and stringent measures to protect sensitive data. In education, integrating LLMs as supplemental tools rather than standalone evaluators could ensure that human expertise remains central to the learning and assessment process.

## Conclusion

This study demonstrates that while LLMs like ChatGPT4, ChatGPT4o, and Claude3-Opus exhibit robust capabilities in generating and solving oral and maxillofacial surgery questions, their performance is not without limitations. None of the models were able to answer correctly all the questions they generated themselves, highlighting persistent challenges such as AI hallucinations and contextual understanding gaps. The results also emphasize the importance of multimodal inputs, as questions with annotated images achieved higher accuracy rates compared to text-only prompts. Despite these shortcomings, the LLMs showed significant promise in problem-solving, logical consistency, and response fidelity, particularly in structured or numerical contexts. Future research should focus on refining these models through HITL approaches, expanding sample sizes, and exploring their adaptability to dynamic medical knowledge and ethical considerations in clinical practice.

## Supporting information

**S1 File.   Questions and answers generated by large language models.**
(ZIP)

## Author contributions

**Conceptualization:** Bong Chul Kim, Kwang Gi Kim.

**Data curation:** Kyuhyung Kim, Bong Chul Kim.

**Formal analysis:** Kyuhyung Kim, Sae Byeol Mun, Young Jae Kim, Bong Chul Kim, Kwang Gi Kim.

**Funding acquisition:** Bong Chul Kim, Kwang Gi Kim.

**Investigation:** Kyuhyung Kim, Sae Byeol Mun.

**Methodology:** Kyuhyung Kim, Bong Chul Kim.

**Project administration:** Bong Chul Kim, Kwang Gi Kim.

**Resources:** Kyuhyung Kim, Bong Chul Kim.

**Software:** Kyuhyung Kim, Sae Byeol Mun.

**Supervision:** Young Jae Kim, Bong Chul Kim, Kwang Gi Kim.

**Validation:** Kyuhyung Kim, Sae Byeol Mun.

**Writing – original draft:** Kyuhyung Kim, Sae Byeol Mun.

**Writing – review & editing:** Young Jae Kim, Bong Chul Kim, Kwang Gi Kim.

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
