## [Decision Letter · Decision Letter 0]

30 Dec 2024

PONE-D-24-47723How valuable are the questions and answers generated by large language models in oral and maxillofacial surgery?PLOS ONE

Dear Dr. KIM,

Thank you for submitting your manuscript to PLOS ONE. After careful consideration, we feel that it has merit but does not fully meet PLOS ONE’s publication criteria as it currently stands. Therefore, we invite you to submit a revised version of the manuscript that addresses the points raised during the review process.

We look forward to receiving your revised manuscript.

Kind regards,

Bekalu Tadesse Moges

Academic Editor

PLOS ONE

Reviewers' comments:

Reviewer's Responses to Questions

**Comments to the Author**

1. Is the manuscript technically sound, and do the data support the conclusions?

Reviewer #1: Partly

Reviewer #2: Partly

Reviewer #3: Partly

Reviewer #4: Yes

Reviewer #5: Yes

2. Has the statistical analysis been performed appropriately and rigorously? 

Reviewer #1: Yes

Reviewer #2: Yes

Reviewer #3: No

Reviewer #4: Yes

Reviewer #5: Yes

3. Have the authors made all data underlying the findings in their manuscript fully available?

Reviewer #1: Yes

Reviewer #2: Yes

Reviewer #3: Yes

Reviewer #4: Yes

Reviewer #5: Yes

4. Is the manuscript presented in an intelligible fashion and written in standard English?

Reviewer #1: Yes

Reviewer #2: Yes

Reviewer #3: Yes

Reviewer #4: Yes

Reviewer #5: Yes

5. Review Comments to the Author

Reviewer #1: General Comments

This paper presents an empirical study investigating the ability of proprietary multimodal large language models to generate and answer questions relevant to oral and maxillofacial surgery. The research topic is both pertinent and timely, and the experiments are conducted with a commendable level of clarity and transparency. The writing is also very clear. However, there are areas where the manuscript can be strengthened, particularly in enhancing the conclusions and addressing ambiguities in the methodology.

Major Comments

- Abstract: The conclusion presented in the abstract is underdeveloped and lacks depth. A more detailed and insightful summary would help emphasize the key findings and contributions of the study.

- Material and methods: The use of open-source images for questions that require visual input is unclear. Since the LLMs used in this study produce only textual output, it is difficult to understand how and why the LLM asked questions requiring an image to answer. Also, how were the appropriate images matched to the questions? Including an example would greatly benefit the readers and clarify this process.

- Material and methods: Intra-rater reliability is mentioned as a metric, but it is unclear how this was evaluated. Did you include duplicate questions during the manual evaluation phase to test for consistency? The results mention that 150 questions were evaluated, meaning this seems unlikely, which complicates the assessment of intra-rater reliability. Or maybe, did you mean you assessed the annotator’s annotation consistency against a final gold standard created after the evaluations by the two surgeons? Either way, this aspect requires clarification.

- Results: The phrase “ground truth labeled by experts” is somewhat misleading. My understanding is that the initial labeling was performed by the LLMs during question generation, with subsequent validation by the two experts. This distinction should be clarified to avoid confusion.

- Conclusion: Similar to the abstract, the conclusion section lacks depth and does not adequately communicate the insights or implications of the study. This is particularly concerning as many readers rely on the abstract and conclusion to determine the value of a paper. Strengthening this section with key lessons and recommendations would significantly improve the manuscript.

Minor Comments

- Abstract: “All three models failed to answer 100% of the questions they created themselves” – I think this sentence can be clearer, I had to check the results to ensure what this is implied. I suggest “None of the three models were able to answer correctly all the questions they generated themselves”.

- Material and methods: Adding a figure to illustrate the prompt used for question generation would be great as it would enhance understanding and improve the transparency of the methodology.

- Discussion: “logical analysis,” or ‘the creation of situations or problems.’ include punctuation within quotation marks. This is a characteristic of LLM-generated contents. It is perfectly fine to use it to improve the phrasing of the sentences, but it would be preferable to adopt a consistent and appropriate style for academic writing by placing punctuation outside quotation marks.

- Table 1: I think it would be great if you could add the % numbers in the non-total values as well.

- Table 4: The clear meaning of “answer” and “basis” is required and should be added in the caption.

Reviewer #2: Manuscript title:

How valuable are the questions and answers generated by large language models in oral and maxillofacial surgery?

The authors aim in this study to evaluate the ability of large language models (LLM) to generate questions and answers in oral and maxillofacial surgery.

The idea is not novel and the rationale behind this repeated aim done by many authors previously is not obvious to me.

This study has so many limitations that could be addressed by expanding this work and generating a more concise reliable conclusion.

Reviewer #3: The study investigates the ability of large language models, including ChatGPT4, ChatGPT4o, and Claude3-Opus, to generate and answer questions related to oral and maxillofacial surgery. The topic is highly relevant and timely, given the growing interest in applying AI in the medical field. While the study addresses an important and emerging area of research, several critical areas need to be improved to enhance the rigor, clarity, and applicability of the findings. Here are some suggestions to improve the manuscript and make it fit for publication:

1. Define the inclusion criteria for the generated questions more clearly, and provide operational definitions for terms like "simple knowledge measurement" and "causal inference."

2. Include real-world clinical scenarios to test the practical relevance of LLM outputs.

3. Use standardized and validated datasets for images instead of open-source images from Google to improve consistency and reliability.

4. Expand the role of oral and maxillofacial surgeons to include input during question generation to ensure clinical relevance and alignment with standardized formats.

5. Evaluate the models using external or independently crafted questions to reduce bias and improve generalizability.

6. Provide detailed descriptions of the statistical methods used, including assumptions, inter-rater reliability, and adjustments for multiple comparisons.

7. Discuss the limitations more explicitly, particularly the reliance on AI-generated content, and its implications for real-world clinical and educational applications.

8. Discuss ethical considerations and propose safeguards to mitigate risks associated with LLM deployment in clinical and educational contexts.

Reviewer #4: Dear Authors,

It was a pleasure to read your manuscript. The study is well-designed and presents valuable insights into the role of large language models in medical education. I believe it is a strong candidate for publication after addressing a few minor comments below.

1.In the introduction, please expand on the rationale for focusing on oral and maxillofacial surgery. Why is this field particularly suited for evaluating large language models compared to other medical specialties? Adding this explanation will strengthen the introduction.

2.In the methodology, please include a clear statement about when the study was conducted. For example, "The study was conducted between [start date] and [end date]." Alternatively, you may specify, "The models were tested using updates from May 16, 2024 (ChatGPT4o), and February 06, 2024 (Claude3-Opus)." This detail will improve transparency. The methodology would also benefit from more details on how the questions were reviewed and validated. Please describe the preprocessing steps used for the data and explain how human reviewers assessed the quality of the generated questions (e.g., the criteria for determining their appropriateness). Additionally, while the use of open-source images is noted, there is no information about the selection process. Please clarify the criteria for choosing the images, how their quality and relevance were ensured, and the process of integrating these images into the study.

3.The results section includes the claim that "all models inferred answers without logical errors," which seems too strong given the variability in fidelity rates. Please revise this to reflect the observed limitations in logical deduction accuracy.

This is an excellent study with strong potential for publication. Addressing these small points will make the manuscript even clearer and more robust. Thank you for your hard work and for sharing this important research.

Reviewer #5: Seeing this groundbreaking and superb work that links the application of cutting-edge IT technologies for human health excites me greatly. It's a very outstanding piece of scientific work. I honestly believe that it deserves to be published.

The research was done in a really good manner. The introduction, abstract, and discussion sections are all greatly interesting.

I only have a few small things to say.

You may review the font size, spacing, and typography of the whole text and make revisions in accordance with the journal's guidelines.

It is understandable that the research did not involve human subjects. The following topics in the methodology and materials section, however, require further clarification.

= When and where was the study carried out?

= How was the analysis carried out?

= Measurements, experimental procedures,

= Who were the specialists who assessed the 150 questions that were produced? How did they get chosen?

The conclusion is extremely short. The conclusions section should be well-written and provide closure because it is the last section in which the findings of this great work are summed up.

6. PLOS authors have the option to publish the peer review history of their article (what does this mean?). If published, this will include your full peer review and any attached files.

Reviewer #1: **Yes: **Jamil Zaghir

Reviewer #2: No

Reviewer #3: No

Reviewer #4: No

Reviewer #5: No

---

## [Author Response · Author response to Decision Letter 1]

28 Jan 2025

Reviewer #1: General Comments

This paper presents an empirical study investigating the ability of proprietary multimodal large language models to generate and answer questions relevant to oral and maxillofacial surgery. The research topic is both pertinent and timely, and the experiments are conducted with a commendable level of clarity and transparency. The writing is also very clear. However, there are areas where the manuscript can be strengthened, particularly in enhancing the conclusions and addressing ambiguities in the methodology.

Major Comments

- Abstract: The conclusion presented in the abstract is underdeveloped and lacks depth. A more detailed and insightful summary would help emphasize the key findings and contributions of the study.

>> As you said, I modified the abstract as follows:

This study demonstrates that while LLMs like ChatGPT4, ChatGPT4o, and Claude3-Opus exhibit robust capabilities in generating and solving oral and maxillofacial surgery questions, their performance is not without limitations. None of the models were able to answer correctly all the questions they generated themselves, highlighting persistent challenges such as AI hallucinations and contextual understanding gaps. The results also emphasize the importance of multimodal inputs, as questions with annotated images achieved higher accuracy rates compared to text-only prompts. Despite these shortcomings, the LLMs showed significant promise in problem-solving, logical consistency, and response fidelity, particularly in structured or numerical contexts.

- Material and methods: The use of open-source images for questions that require visual input is unclear. Since the LLMs used in this study produce only textual output, it is difficult to understand how and why the LLM asked questions requiring an image to answer. Also, how were the appropriate images matched to the questions? Including an example would greatly benefit the readers and clarify this process.

>> Interpreting photography, X-rays, CT scans and/or MRIs is an important task in most fields of medicine, including oral and maxillofacial surgery. So, we asked LLM questions requiring an image to answer, even though we knew that LLM produces only textual output.

And we modified the methods as follows:

This study evaluated three LLMs: ChatGPT4, ChatGPT4o, and Claude3-Opus. The models were tested using updates from May 16, 2024 (ChatGPT4 and ChatGPT4o), and February 06, 2024 (Claude3-Opus). This study was conducted between June 12, 2024, and July 10, 2024. We asked three LLMs the following questions (Fig. 1):

Create 50 questions for the Oral and Maxillofacial Surgery Specialist Exam to find 1 correct answer out of 5 options and provide the correct answer for each question and the basis for the correct answer.

Each question must meet one or more of the following:

1. Simple knowledge measurement

2. Causal inference

3. Example through patient case

4. Includes schematic diagram, photography, X-ray, CT and/or MRI

Two oral and maxillofacial surgeons reviewed the questions generated by each LLM. To ensure proper preprocessing of the data, all generated questions were systematically checked for compliance with the predefined criteria (knowledge measurement, causal inference, patient case examples, and inclusion of visual aids). They confirmed that the generated questions were appropriate for oral and maxillofacial surgery. And they assessed the quality of questions by evaluating their clinical relevance, alignment with standardized exam formats, and logical flow of answers and explanations. Criteria included the accuracy of answers, the thoroughness of the problem-solving process, and the logical consistency of explanations, with inconsistencies flagged for further analysis. Furthermore, each question was categorized into specific fields (e.g., dentoalveolar, trauma, infection, cyst, etc.).

Out of the 150 questions generated by three LLMs, open-source images from Google were inserted for those requiring images (ChatGPT4: 9 questions, ChatGPT4o: 18 questions, Claude3-opus: 18 questions). We built standardized and validated datasets for images using only those from credible institutions among open sources. For questions where only a part of the image was to be evaluated, the relevant part was highlighted using shapes like arrows or circles (Fig. 2). Answers and explanations for each question were removed and then provided to the three LLMs to generate new answers and explanations.

And we added the following figures:

Figure 1. A flow diagram showing the process by which large language models generate and answer questions about oral and maxillofacial surgery in this study. LLM: large language models, OMFS: oral and maxillofacial surgeon.

Figure 2. An example of inserting images verified and labeled by experts from credible institutions among open sources into questions generated by large language models.

- Material and methods: Intra-rater reliability is mentioned as a metric, but it is unclear how this was evaluated. Did you include duplicate questions during the manual evaluation phase to test for consistency? The results mention that 150 questions were evaluated, meaning this seems unlikely, which complicates the assessment of intra-rater reliability. Or maybe, did you mean you assessed the annotator’s annotation consistency against a final gold standard created after the evaluations by the two surgeons? Either way, this aspect requires clarification.

>> I agree with your comment and have revised it as follows:

Evaluations recorded results as correct or incorrect and classified explanations as detailed or insufficient. Image interpretation ability was assessed, and answers were cross verified to evaluate the problem-solving abilities and logical consistency of each model. Intra-rater and Inter-rater reliability were evaluated for a total of 150 by two oral and maxillofacial surgeons. Each author evaluated a total of 150 questions at 1-week intervals. We assessed the annotator’s annotation consistency against a final gold standard created after the evaluations by the two surgeons.

- Results: The phrase “ground truth labeled by experts” is somewhat misleading. My understanding is that the initial labeling was performed by the LLMs during question generation, with subsequent validation by the two experts. This distinction should be clarified to avoid confusion.

>> I agree with your comment and have revised it as follows:

A confusion matrix was calculated using the ground truth (initial labeling performed by the LLMs during question generation, with subsequent validation by the two experts) for each model's questions and responses (Fig. 4).

- Conclusion: Similar to the abstract, the conclusion section lacks depth and does not adequately communicate the insights or implications of the study. This is particularly concerning as many readers rely on the abstract and conclusion to determine the value of a paper. Strengthening this section with key lessons and recommendations would significantly improve the manuscript.

>> I totally agree with your opinion. So, I revised conclusion as follows:

This study demonstrates that while LLMs like ChatGPT4, ChatGPT4o, and Claude3-Opus exhibit robust capabilities in generating and solving oral and maxillofacial surgery questions, their performance is not without limitations. None of the models were able to answer correctly all the questions they generated themselves, highlighting persistent challenges such as AI hallucinations and contextual understanding gaps. The results also emphasize the importance of multimodal inputs, as questions with annotated images achieved higher accuracy rates compared to text-only prompts. Despite these shortcomings, the LLMs showed significant promise in problem-solving, logical consistency, and response fidelity, particularly in structured or numerical contexts. Future research should focus on refining these models through HITL approaches, expanding sample sizes, and exploring their adaptability to dynamic medical knowledge and ethical considerations in clinical practice.

Minor Comments

- Abstract: “All three models failed to answer 100% of the questions they created themselves” – I think this sentence can be clearer, I had to check the results to ensure what this is implied. I suggest “None of the three models were able to answer correctly all the questions they generated themselves”.

>> As you said, I modified the abstract as follows:

Each model exhibited a correct answer rate of over 90%. None of the three models were able to answer correctly all the questions they generated themselves.

- Material and methods: Adding a figure to illustrate the prompt used for question generation would be great as it would enhance understanding and improve the transparency of the methodology.

>> As replied to previous comment, we added figures.

- Discussion: “logical analysis,” or ‘the creation of situations or problems.’ include punctuation within quotation marks. This is a characteristic of LLM-generated contents. It is perfectly fine to use it to improve the phrasing of the sentences, but it would be preferable to adopt a consistent and appropriate style for academic writing by placing punctuation outside quotation marks.

>> I modified the sentence so that the punctuation is outside the quotation marks, such as "logical analysis", or 'the creation of situations or problems'.

- Table 1: I think it would be great if you could add the % numbers in the non-total values as well.

>> I added the % numbers in the non-total values as well.

Generated by ChatGPT4

(n=50) Generated by ChatGPT4o

(n=50) Generated by Claude3-Opus

(n=50) Total (response rate)

Answered by ChatGPT4 44 (88.0%) 46 (92.0%) 47 (94.0%) 137 (91.3 %)

Answered by ChatGPT4o 44 (88.0%) 47 (94.0%) 44 (88.0%) 135 (90.0 %)

Answered by Claude3-Opus 43 (86.0%) 47 (94.0%) 49 (98.0%) 139 (92.7 %)

Total (response rate) 131 (87.3%) 140 (93.3%) 140 (93.3%)

- Table 4: The clear meaning of “answer” and “basis” is required and should be added in the caption.

>> I added the following to the caption:

answer: the answer generated by each model for the input question.

basis: description of the basis on which each model generated the answer.

Reviewer #2: Manuscript title:

How valuable are the questions and answers generated by large language models in oral and maxillofacial surgery?

The authors aim in this study to evaluate the ability of large language models (LLM) to generate questions and answers in oral and maxillofacial surgery.

The idea is not novel and the rationale behind this repeated aim done by many authors previously is not obvious to me.

This study has so many limitations that could be addressed by expanding this work and generating a more concise reliable conclusion.

>> Thank you for taking the time to review this manuscript. We have thoroughly revised this manuscript based on the comments of other reviewers. If you don't mind, I would appreciate it if you could review the revised manuscript again.

Reviewer #3: The study investigates the ability of large language models, including ChatGPT4, ChatGPT4o, and Claude3-Opus, to generate and answer questions related to oral and maxillofacial surgery. The topic is highly relevant and timely, given the growing interest in applying AI in the medical field. While the study addresses an important and emerging area of research, several critical areas need to be improved to enhance the rigor, clarity, and applicability of the findings. Here are some suggestions to improve the manuscript and make it fit for publication:

1. Define the inclusion criteria for the generated questions more clearly, and provide operational definitions for terms like "simple knowledge measurement" and "causal inference."

>> Based on your comments, we have added the following to the discussion:

In this study, we presented several conditions when each LLM generated questions. The "simple knowledge measurement" was intended to measure each model’s understanding or recall of basic concepts, facts, or information. “Causal inference” was to confirm the LLM's process of determining whether and how one variable directly influences another through evidence, reasoning, or statistical analysis. “Example through patient case” was intended to confirm LLMs’ ability to create a detailed narrative or scenario describing a specific patient’s condition, history, and treatment to illustrate medical principles or practices. And "includes schematic diagram, photography, X-ray, CT, and/or MRI" was to evaluate the LLMs' understanding of anatomical or physiological features through medical image data. As a result, we enabled LLMs to generate questions appropriate to the purpose of this study.

2. Include real-world clinical scenarios to test the practical relevance of LLM outputs.

>> Good point. However, in this study, the conditions for generating questions in LLMs were presented as "example through patient case" and "includes schematic diagram, photography, X-ray, CT and/or MRI". We believe that these conditions are consistent with real-world clinical scenarios to test the practical relevance of LLM outputs. However, since your point is valid, we have revised the discussion as follows:

Future research could expand the scope to include a wider range of surgical specialties, evaluate real-world clinical scenarios to test the practical relevance of LLM outputs, use standardized medical images, and assess the models' ability to incorporate real-time updates in medical practice. Evaluating patient-centered decision-making capabilities and ethical considerations related to AI-generated content could also provide valuable insights.

3. Use standardized and validated datasets for images instead of open-source images from Google to improve consistency and reliability.

>> Good point. However, this study aimed to evaluate the ability of LLMs to generate answers and questions in oral and maxillofacial surgery. And the creation principle of LLMs is based on learning the patterns and structures of language by utilizing deep learning technology and large-scale text data. In this process, they collect large amounts of text data from various sources such as web documents, books, papers, and code. For the evaluation of these LLMs, we felt it would be more appropriate to use open-source images from Google rather than our hospital's patient data. Among the open-source images, only those from credible institutions that two oral and maxillofacial surgeons deemed appropriate were used.

And we modified our methodology as follows:

Out of the 150 questions generated by three LLMs, open-source images from Google were inserted for those requiring images (ChatGPT4: 9 questions, ChatGPT4o: 18 questions, Claude3-opus: 18 questions). We built standardized and validated datasets for images using only those from credible institutions among open sources. For questions where only a part of the image was to be evaluated, the relevant part was highlighted using shapes like arrows or circles (Fig. 2).

4. Expand the role of oral and maxillofacial surgeons to include input during question generation to ensure clinical relevance and alignment with standardized formats.

>> During the implementation phase of this study, we verified the clinical relevance and alignment with standardized formats of the questions and answers generated by LLMs. Based on your comments, we modified the methodology as follows:

Two oral and maxillofacial surgeons reviewed the questions generated by each LLM. To ensure proper preprocessing of the data, all generated questions were systematically checked for compliance with the predefined criteria (knowledge measurement, causal inference, patient case examples, and inclusion of visual aids). They confirmed that the generated questions were appropriate for oral and maxillofacial surgery. And they assessed the quality of questions by evaluating their clinical relevance, alignment with standardized exam formats, and logical flow of answers and explanations. Criteria included the accuracy of answers, the thoroughness of the problem-solving process, and the logical consistency of explanations, with inconsistencies flagged for further analysis. Furthermore, each

---

## [Decision Letter · Decision Letter 1]

20 Feb 2025

PONE-D-24-47723R1How valuable are the questions and answers generated by large language models in oral and maxillofacial surgery?PLOS ONE

Dear Dr. KIM,

Thank you for submitting your manuscript to PLOS ONE. After careful consideration, we feel that it has merit but does not fully meet PLOS ONE’s publication criteria as it currently stands. Therefore, we invite you to submit a revised version of the manuscript that addresses the points raised during the review process.

Dear Authors, Please address the concerns of the reviewer or justify why those concerns are not addressed in the previous revision.

We look forward to receiving your revised manuscript.

Kind regards,

Bekalu Tadesse Moges

Academic Editor

PLOS ONE

Journal Requirements:

Reviewers' comments:

Reviewer's Responses to Questions

**Comments to the Author**

1. If the authors have adequately addressed your comments raised in a previous round of review and you feel that this manuscript is now acceptable for publication, you may indicate that here to bypass the “Comments to the Author” section, enter your conflict of interest statement in the “Confidential to Editor” section, and submit your "Accept" recommendation.

Reviewer #2: All comments have been addressed

Reviewer #5: (No Response)

2. Is the manuscript technically sound, and do the data support the conclusions?

Reviewer #2: Yes

Reviewer #5: Yes

3. Has the statistical analysis been performed appropriately and rigorously? 

Reviewer #2: Yes

Reviewer #5: Yes

4. Have the authors made all data underlying the findings in their manuscript fully available?

Reviewer #2: Yes

Reviewer #5: Yes

5. Is the manuscript presented in an intelligible fashion and written in standard English?

Reviewer #2: Yes

Reviewer #5: Yes

6. Review Comments to the Author

Reviewer #2: Thank you for addressing the reviewers comments which resulted in the improvement of the manuscript.

Reviewer #5: As I said in the first review, the work of is of great value. It is well done work. The few minor comments that I raised were not done. There is also no response to the reviewers' comments.

7. PLOS authors have the option to publish the peer review history of their article (what does this mean?). If published, this will include your full peer review and any attached files.

Reviewer #2: No

Reviewer #5: No

---

## [Author Response · Author response to Decision Letter 2]

23 Feb 2025

Reviewer #2:

Thank you for addressing the reviewers comments which resulted in the improvement of the manuscript.

>> Thank you very much. Your reviews have greatly improved this manuscript.

Reviewer #5:

As I said in the first review, the work of is of great value. It is well done work. The few minor comments that I raised were not done. There is also no response to the reviewers' comments.

>> Following your comments, we have revisited your comments raised in the previous round.

Seeing this groundbreaking and superb work that links the application of cutting-edge IT technologies for human health excites me greatly. It's a very outstanding piece of scientific work. I honestly believe that it deserves to be published.

The research was done in a really good manner. The introduction, abstract, and discussion sections are all greatly interesting.

I only have a few small things to say.

You may review the font size, spacing, and typography of the whole text and make revisions in accordance with the journal's guidelines.

>> I think this was revised in the previous round.

It is understandable that the research did not involve human subjects. The following topics in the methodology and materials section, however, require further clarification.

>> For these comments, I revised the methods as follows:

= When and where was the study carried out?

This study evaluated three LLMs: ChatGPT4, ChatGPT4o, and Claude3-Opus. The models were tested using updates from May 16, 2024 (ChatGPT4 and ChatGPT4o), and February 06, 2024 (Claude3-Opus). This study was conducted between June 12, 2024, and July 10, 2024, at Wonkwang and Gachon University, Republic of Korea.

= How was the analysis carried out?

Question Generation and Evaluation

Three LLMs were prompted with the following request (Fig. 1):

Create 50 questions for the Oral and Maxillofacial Surgery Specialist Exam to find 1 correct answer out of 5 options and provide the correct answer for each question and the basis for the correct answer.

Each question must meet one or more of the following:

1. Simple knowledge measurement

2. Causal inference

3. Example through patient case

4. Includes schematic diagram, photography, X-ray, CT and/or MRI

Image Integration and Modification

For questions requiring images, open-source images from Google were incorporated, ensuring they were sourced only from credible institutions. Image utilization per model was as follows: ChatGPT-4 (9 questions), ChatGPT-4o (18 questions), and Claude 3-Opus (18 questions). When only a specific part of an image needed to be evaluated, relevant areas were highlighted using arrows or circles for clarity (Fig. 2). To assess LLMs' ability to generate independent explanations, the provided answers and rationales were removed before resubmitting the questions to the models.

= Measurements, experimental procedures,

Performance Evaluation and Statistical Analysis

The LLMs' capabilities were assessed based on:

• Logical Accuracy and Thoroughness: If an explanation contained correct isolated facts but lacked logical relevance to the problem, it was deemed incorrect. Similarly, a correct solution with an explanation consisting only of disjointed facts was considered insufficient.

• Quantitative Assessment: Responses were classified as correct or incorrect, and explanations were labeled as detailed or insufficient. Image interpretation abilities were also evaluated, and cross-verifications were performed to ensure logical consistency.

• Reliability Testing: Intra-rater and inter-rater reliability were assessed for all 150 questions. Each expert evaluated all questions twice at a one-week interval, and a final gold-standard evaluation was established based on consensus.

• Statistical Analysis: An independent t-test was conducted to compare performance metrics among the three LLMs. Accuracy, precision, recall, and F1-scores were calculated using Scikit-learn version 1.4.2. The statistical analyses assumed normality in data distribution and independence between observations. Adjustments for multiple comparisons were considered unnecessary as the analyses were hypothesis-driven and controlled.

= Who were the specialists who assessed the 150 questions that were produced? How did they get chosen?

Expert Review Process

Two oral and maxillofacial surgeons assessed the generated questions. These specialists were selected based on their expertise in oral and maxillofacial surgery and their prior experience with standardized examination question development. The review process involved:

• Verifying compliance with predefined criteria (knowledge measurement, causal inference, patient case-based scenarios, and image inclusion).

• Assessing clinical relevance, alignment with standardized exam formats, and logical coherence of the provided explanations.

• Evaluating the accuracy of answers, completeness of problem-solving approaches, and consistency of logical reasoning.

Categorizing questions into subfields such as dentoalveolar, trauma, infection, and cysts.

The conclusion is extremely short. The conclusions section should be well-written and provide closure because it is the last section in which the findings of this great work are summed up.

>> I think this was revised in the previous round.

---

## [Editor Report · Decision Letter 2]

26 Feb 2025

PONE-D-24-47723R2How valuable are the questions and answers generated by large language models in oral and maxillofacial surgery?PLOS ONE

Dear Dr. KIM,

Thank you for submitting your manuscript to PLOS ONE. After careful consideration, we feel that it has merit but does not fully meet PLOS ONE’s publication criteria as it currently stands. Therefore, we invite you to submit a revised version of the manuscript that addresses the points raised during the review process.

Dear authors, your work is relevant and we appreciate all the efforts you put into it. However, please revising the manuscript considering the valuable comments given by the reviewers (see below), as they are not addressed in the revised version. Particularly pay attention to the methods section, as it should be reproducible to other researchers. this in turn requires detailed discussion of the procedures and methods

Seeing this groundbreaking and superb work that links the application of cutting-edge IT technologies for human health excites me greatly. It's a very outstanding piece of scientific work. I honestly believe that it deserves to be published.

The research was done in a really good manner. The introduction, abstract, and discussion sections are all greatly interesting.

I only have a few small things to say.

You may review the font size, spacing, and typography of the whole text and make revisions in accordance with the journal's guidelines.

It is understandable that the research did not involve human subjects. The following topics in the methodology and materials section, however, require further clarification.

= When and where was the study carried out?

= How was the analysis carried out?

= Measurements, experimental procedures,

= Who were the specialists who assessed the 150 questions that were produced? How did they get chosen?

The conclusion is extremely short. The conclusions section should be well-written and provide closure because it is the last section in which the findings of this great work are summed up.

We look forward to receiving your revised manuscript.

Kind regards,

Bekalu Tadesse Moges

Academic Editor

PLOS ONE
---

## [Author Response · Author response to Decision Letter 3]

2 Mar 2025

Seeing this groundbreaking and superb work that links the application of cutting-edge IT technologies for human health excites me greatly. It's a very outstanding piece of scientific work. I honestly believe that it deserves to be published.

The research was done in a really good manner. The introduction, abstract, and discussion sections are all greatly interesting.

I only have a few small things to say.

You may review the font size, spacing, and typography of the whole text and make revisions in accordance with the journal's guidelines.

>> Based on your feedback, we have revised the font size, spacing, and typography of the whole text in accordance with the journal's guidelines.

It is understandable that the research did not involve human subjects. The following topics in the methodology and materials section, however, require further clarification.

>> To clarify the materials and methods, we added the following figure.

Figure 1. A flow diagram showing the process by which large language models generate and answer questions about oral and maxillofacial surgery in this study. LLM: large language models, OMFS: oral and maxillofacial surgeon.

Figure 2. An example of inserting images verified and labeled by experts from credible institutions among open sources into questions generated by large language models.

= When and where was the study carried out?

>> This study was conducted between June 12, 2024, and July 10, 2024, at Wonkwang and Gachon University, Republic of Korea. We revised the methods as follows:

This study evaluated three LLMs: ChatGPT4, ChatGPT4o, and Claude3-Opus. The models were tested using updates from May 16, 2024 (ChatGPT4 and ChatGPT4o), and February 06, 2024 (Claude3-Opus). This study was conducted between June 12, 2024, and July 10, 2024, at Wonkwang and Gachon University, Republic of Korea.

= How was the analysis carried out?

>> We conducted Performance Evaluation and Statistical Analysis as follows:

The LLMs' capabilities were assessed based on:

• Logical Accuracy and Thoroughness: If an explanation contained correct isolated facts but lacked logical relevance to the problem, it was deemed incorrect. Similarly, a correct solution with an explanation consisting only of disjointed facts was considered insufficient.

• Quantitative Assessment: Responses were classified as correct or incorrect, and explanations were labeled as detailed or insufficient. Image interpretation abilities were also evaluated, and cross-verifications were performed to ensure logical consistency.

• Reliability Testing: Intra-rater and inter-rater reliability were assessed for all 150 questions. Each expert evaluated all questions twice at a one-week interval, and a final gold-standard evaluation was established based on consensus.

• Statistical Analysis: An independent t-test was conducted to compare performance metrics among the three LLMs. Accuracy, precision, recall, and F1-scores were calculated using Scikit-learn version 1.4.2. The statistical analyses assumed normality in data distribution and independence between observations. Adjustments for multiple comparisons were considered unnecessary as the analyses were hypothesis-driven and controlled.

= Measurements, experimental procedures,

>> We had LLMs generate questions and evaluated the questions they generated. For questions that require an image, we found and inserted the appropriate image. And each LLM was asked to solve the questions and explain them appropriately. We revised the methods as follows:

Question Generation and Evaluation

Three LLMs were prompted with the following request (Fig. 1):

Create 50 questions for the Oral and Maxillofacial Surgery Specialist Exam to find 1 correct answer out of 5 options and provide the correct answer for each question and the basis for the correct answer.

Each question must meet one or more of the following:

1. Simple knowledge measurement

2. Causal inference

3. Example through patient case

4. Includes schematic diagram, photography, X-ray, CT and/or MRI

Expert Review Process

Two oral and maxillofacial surgeons assessed the generated questions. These specialists were selected based on their expertise in oral and maxillofacial surgery and their prior experience with standardized examination question development. The review process involved:

• Verifying compliance with predefined criteria (knowledge measurement, causal inference, patient case-based scenarios, and image inclusion).

• Assessing clinical relevance, alignment with standardized exam formats, and logical coherence of the provided explanations.

• Evaluating the accuracy of answers, completeness of problem-solving approaches, and consistency of logical reasoning.

• Categorizing questions into subfields such as dentoalveolar, trauma, infection, and cysts.

Image Integration and Modification

For questions requiring images, open-source images from Google were incorporated, ensuring they were sourced only from credible institutions. Image utilization per model was as follows: ChatGPT-4 (9 questions), ChatGPT-4o (18 questions), and Claude 3-Opus (18 questions). When only a specific part of an image needed to be evaluated, relevant areas were highlighted using arrows or circles for clarity (Fig. 2). To assess LLMs' ability to generate independent explanations, the provided answers and rationales were removed before resubmitting the questions to the models.

= Who were the specialists who assessed the 150 questions that were produced? How did they get chosen?

>> The specialists who assessed the 150 questions that were produced were two oral and maxillofacial surgeons. These specialists were selected based on their expertise in oral and maxillofacial surgery and their prior experience with standardized examination question development. We revised the methods as follows:

Expert Review Process

Two oral and maxillofacial surgeons assessed the generated questions. These specialists were selected based on their expertise in oral and maxillofacial surgery and their prior experience with standardized examination question development. The review process involved:

• Verifying compliance with predefined criteria (knowledge measurement, causal inference, patient case-based scenarios, and image inclusion).

• Assessing clinical relevance, alignment with standardized exam formats, and logical coherence of the provided explanations.

• Evaluating the accuracy of answers, completeness of problem-solving approaches, and consistency of logical reasoning.

• Categorizing questions into subfields such as dentoalveolar, trauma, infection, and cysts.

The conclusion is extremely short. The conclusions section should be well-written and provide closure because it is the last section in which the findings of this great work are summed up.

>> Based on your comments, we have revised the conclusion as follows:

This study demonstrates that while LLMs like ChatGPT4, ChatGPT4o, and Claude3-Opus exhibit robust capabilities in generating and solving oral and maxillofacial surgery questions, their performance is not without limitations. None of the models were able to answer correctly all the questions they generated themselves, highlighting persistent challenges such as AI hallucinations and contextual understanding gaps. The results also emphasize the importance of multimodal inputs, as questions with annotated images achieved higher accuracy rates compared to text-only prompts. Despite these shortcomings, the LLMs showed significant promise in problem-solving, logical consistency, and response fidelity, particularly in structured or numerical contexts. Future research should focus on refining these models through HITL approaches, expanding sample sizes, and exploring their adaptability to dynamic medical knowledge and ethical considerations in clinical practice.

---

## [Editor Report · Decision Letter 3]

24 Mar 2025

How valuable are the questions and answers generated by large language models in oral and maxillofacial surgery?

PONE-D-24-47723R3

Dear Dr. Bong Chul KIM,

We’re pleased to inform you that your manuscript has been judged scientifically suitable for publication and will be formally accepted for publication once it meets all outstanding technical requirements.

Kind regards,

Bekalu Tadesse Moges

Academic Editor

PLOS ONE
---

## [Editor Report · Acceptance letter]

PONE-D-24-47723R3

PLOS ONE

Dear Dr. Kim,

I'm pleased to inform you that your manuscript has been deemed suitable for publication in PLOS ONE. Congratulations! Your manuscript is now being handed over to our production team.

Kind regards,

on behalf of

Dr. Bekalu Tadesse Moges

Academic Editor

PLOS ONE